# Evaluation of Two Simultaneous Metabolomic and Proteomic Extraction Protocols Assessed by Ultra-High-Performance Liquid Chromatography Tandem Mass Spectrometry

**DOI:** 10.3390/ijms24021354

**Published:** 2023-01-10

**Authors:** Ruba A. Zenati, Alexander D. Giddey, Hamza M. Al-Hroub, Yousra A. Hagyousif, Waseem El-Huneidi, Yasser Bustanji, Eman Abu-Gharbieh, Mohammad A. Y. Alqudah, Mohd Shara, Ahmad Y. Abuhelwa, Nelson C. Soares, Mohammad H. Semreen

**Affiliations:** 1Department of Medicinal Chemistry, College of Pharmacy, University of Sharjah, Sharjah 27272, United Arab Emirates; 2Research Institute of Medical and Health Sciences, University of Sharjah, Sharjah 27272, United Arab Emirates; 3Department of Basic Medical Sciences, College of Medicine, University of Sharjah, Sharjah 27272, United Arab Emirates; 4Department of Basic and Clinical Pharmacology, College of Medicine, University of Sharjah, Sharjah 27272, United Arab Emirates; 5School of Pharmacy, The University of Jordan, Amman 11942, Jordan; 6Department of Clinical Sciences, College of Medicine, University of Sharjah, Sharjah 27272, United Arab Emirates; 7Department of Pharmacy Practice and Pharmacotherapeutics, College of Pharmacy, University of Sharjah, Sharjah 27272, United Arab Emirates; 8Department of Clinical Pharmacy, Faculty of Pharmacy, Jordan University of Science and Technology, Irbid 22110, Jordan; 9Department of Human Genetics, National Institute of Health Doutor Ricardo Jorge (INSA), 1649-016 Lisbon, Portugal

**Keywords:** metabolomics, proteomics, multi-omics, plasma, reproducibility

## Abstract

Untargeted multi-omics analysis of plasma is an emerging tool for the identification of novel biomarkers for evaluating disease prognosis, and for developing a better understanding of molecular mechanisms underlying human disease. The successful application of metabolomic and proteomic approaches relies on reproducibly quantifying a wide range of metabolites and proteins. Herein, we report the results of untargeted metabolomic and proteomic analyses from blood plasma samples following analyte extraction by two frequently-used solvent systems: chloroform/methanol and methanol-only. Whole blood samples were collected from participants (n = 6) at University Hospital Sharjah (UHS) hospital, then plasma was separated and extracted by two methods: (i) methanol precipitation and (ii) 4:3 methanol:chloroform extraction. The coverage and reproducibility of the two methods were assessed by ultra-high-performance liquid chromatography-electrospray ionization quadrupole time-of-flight mass spectrometry (UHPLC-ESI-QTOF-MS). The study revealed that metabolite extraction by methanol-only showed greater reproducibility for both metabolomic and proteomic quantifications than did methanol/chloroform, while yielding similar peptide coverage. However, coverage of extracted metabolites was higher with the methanol/chloroform precipitation.

## 1. Introduction

Omics studies, including genomics, epigenomics, transcriptomics, proteomics, and metabolomics, have advanced the early diagnosis of diseases [1]. Each of these ‘omics technologies provides data on hundreds or even thousands of analytes and the potential characteristic quantitative differences between disease states that can be utilized as markers of underlying disease processes. Individually, these technologies have provided sufficiently wide coverage to allow for successful investigation into which biological pathways are affected in various disease states [2,3]. However, the over-dependence on any single ‘omics technology will necessarily limit the available information regarding the cause of these changes. Hence, it is becoming increasingly important to combine multiple layers of ‘omics data when assessing the underlying causative changes that lead to disease [4,5]. Furthermore, there have been several strategies developed to determine the relationship between proteomic and metabolomic data from the same samples [6,7,8,9,10].

Blood plasma-based multi-omics research presently dominates the field due to its less-invasive sampling methods and the nature of blood in that it reflects the global metabolic response to many stimuli, such as disease development, applied medication or nutrition, genetic change, and environmental variables [11,12]. For multi-omics studies, the sample preparation process should be reproducible, simple, fast, and unbiased [13]. A well-selected and optimized sample preparation method is crucial for the accurate assessment of the metabolic and proteomic profile, and for the biological interpretation of the findings [14].

The most-widely-used sample preparation techniques for LC-MS-based plasma untargeted metabolomics and proteomics typically use protein precipitation by different organic solvents—including methanol [15], butanol [16], methyl-t-butyl ether (MTBE) [17,18], chloroform [19], and ethylacetate [20], among others—or a combination of them, followed by centrifugation and/or supernatant filtering. These processes each vary with respect to metabolomic coverage and reproducibility [21]. Specifically, one of the most commonly-used single-phase extraction solvents for plasma metabolomics is methanol-based protein precipitation [22,23,24]. Less hydrophobic solvents take into consideration the hydrophilic environment that plasma represents, but restrict the analysis of hydrophobic metabolites such as lipids, fatty acids, steroids, and thyroid hormones, which are often transported via carrier proteins in the blood [25]. A limitation is the co-precipitation of plasma metabolites during the protein precipitation step which results in lowered corresponding metabolite intensities, a particular concern in disease-related metabolomic analyses [26]. 

Additionally, the various solvent extraction systems differ in the number of phases, with uniphasic solutions such as methanol-only, and multiphasic solutions such as that obtained with the MTBE or the long-standing chloroform/methanol method. Biphasic separation techniques are used in the field of multi-omics, the motivation initially being to remove potentially interfering compounds from the downstream analytical method to improve analysis quality, and it was initially designed to extract either pure lipids or polar metabolites [27]. The most common example is the use of the chloroform-methanol extraction process, where the resulting two phases are subsequently combined to gather both liquid phases [15,28]. Unfortunately, the primary limitation that is associated with biphasic protocols, such as with the chloroform-based extraction, is that reproducibility is lowered because of the amorphous, solid components interphase which is located between the two immiscible layers. As a result, this diffuse interphase will make it difficult to collect this insoluble fraction quantitatively [29].

Due to the above-mentioned specificity of each extraction, multiple aliquots of the same sample are usually required to obtain sufficient material to be analyzed accurately. In addition to the increased effort caused by multiple parallel sample handling, separate extraction protocols per sample also necessitate the use of more sample material, which may be limiting many cases, such as when sample types are rare. Consequently, a reliable and robust extraction method is required to recover the major molecular component from a single sample aliquot. Thus, in this study, we compared the coverage and reproducibility of two extraction methods commonly employed in plasma studies: the single-phase methanol precipitation (MeOH ppt), and biphasic chloroform/methanol (MeOH:CHCl_3_) extraction, though untargeted metabolomics and proteomics analysis using the identical plasma samples.

## 2. Results

We compared two commonly-employed simultaneous metabolome and proteome extraction methods for blood plasma by LC-MS/MS from samples derived from multiple individuals, incorporating biological variation, and repeated analysis of aliquots from a single sample, thus examining technical variation of the two protocols. As each protocol yields both a metabolomic and proteomic fraction, the coverage and overlap of identified metabolites and peptides were compared as well as the variability of quantification. 

### 2.1. Biological Metabolomic Variation Exceeds That of Both Protocols

To explore the variations between MeOH ppt and MeOH:CHCl_3_ extractions, we evaluated the coverage and reproducibility in the metabolomics of six different plasma samples collected from healthy individuals by both extraction methods.

By way of coverage, most (74, see Figure 1B) of the metabolites identified were recovered by both extraction methods. However, there were 9 identified metabolites observed only in the MeOH:CHCl_3_ extraction. Specifically, these were saccharopine, glycylproline, rhamnose, N-acetyl-L-tyrosine, heparin, butanone, pregnenolone sulfate, acetic acid and 2,2-dimethylsuccinic acid. Additionally, of the 74 metabolites recovered by both methods, there were 10 metabolites (isobutyric acid, 2-phenylaminoadenosine, diacetyl, cinnamaldehyde, succinylacetone, 2-Ketohexanoic acid, 3,4-dihydroxyphenylglycol, glycerol, acetaldehyde and 3,4,5-trimethoxycinnamic acid) present with higher intensities (*p* < 0.05, Fold-change > 2) in the MeOH:CHCl_3_ extraction and 8 (pyridine, pipecolic acid, L-lysine, L-proline, L-glutamine, benzamide, linoleic acid, and L-valine) with improved recovery in the MeOH-only extraction (see Figure 1C).

In evaluating the quantitative variability of each method, we compared the coefficient of variance (standard deviation divided by the mean, CV) for each identified metabolite for each method and compared the distributions of the CV values between the two methods. Figure 1A shows that not statistically significant (*p*-value 0.451926) difference could be observed when comparing the two methods across multiple samples sourced from different individuals. While the observed CV values trended lower in MeOH ppt than MeOH:CHCl_3_ extraction across the six different plasma samples (mean CV of 0.389594 for MeOH ppt in comparison to MeOH:CHCl_3_ extraction 0.427405), if any true difference existed, it was exceeded by the inherent biological variation between the samples.

#### Technical Variation Is Improved in MeOH-Only Extraction

Variation in biological replicates includes both biological and technical variation, so biological variability may mask technical reproducibility. Thus, we sought to compare the reproducibility of the technical protocols more precisely by considering quadruplicate extractions for each extraction method from aliquots of a single biological sample.

Similarly to that observed above, Figure 2B shows that there were 11 of 83 metabolites exclusively identified from the MeOH:CHCl_3_ extracts, including saccharopine, 2-ketohexanoic acid, glycylproline, rhamnose, N-acetyl-L-tyrosine, heparin, butanone, pregnenolone sulfate, acetic acid, 2-phenylaminoadenosine and diacetyl. An additional 3 (3,4-dihydroxyphenylglycol, glycerol and 3,4,5-trimethoxycinnamic acid) were recovered with significantly higher intensities from the MeOH:CHCl_3_ extracts (see Figure 2C). Meanwhile, 5 metabolites (pyridine, pipecolic acid, L-lysine, L-proline and benzamide) were recovered with significantly greater intensities from the MeOH-only extraction. 

Figure 2A shows that, unlike that seen with the biological replicates, the variation observed for identified metabolites was ~50% greater (0.275 vs. 0.179, *p* < 0.01) with MeOH:CHCl_3_ extracts when compared to that seen in MeOH-only extracts. Thus, when considering only the merits of the respective protocols and not the variability inherent to the samples, it appears that the MeOH-only extraction shows improved reproducibility with respect to quantitation, while MeOH:CHCl_3_ extraction appears to yield generally higher recoveries of metabolites, with the same trend being observed with respect to unidentified analyte features, where over 5500 molecular features were detected by the MetaboScape software, as contrasted with >3500 such features in the MeOH-only extraction (see Table 1).

### 2.2. Proteomics Analysis

Continuing our technical evaluation of the two methods, we assessed the coverage and reproducibility of the proteome extracted from the same biological sample used for the technical evaluation of metabolomics (see above). Six aliquots (three for each method) from the same biological sample were extracted by either methanol (MeOH) ppt or MeOH:CHCl_3_, and proteins were then digested and prepared for LC-MS/MS as described above. The data were processed using the commonly-used proteomics data processing tool MaxQuant [30], which includes the popular label-free quantitation (LFQ) algorithm MaxLFQ. We compared the reproducibility of raw peptide intensities, raw (summed peptides) protein intensities, and protein LFQ values.

Figure 3 shows that the quantitative variation, as assessed by the distribution of CV values, was significantly (*p* < 0.0001) less in MeOH-only protein precipitation than was observed from MeOH:CHCl_3_ precipitation for raw intensity values at both the peptide (0.369 vs. 0.483) and protein levels (0.265 vs. 0.482). However, it was noteworthy that when using the MaxLFQ algorithm for protein quantitation, the variation for protein quantitation was improved for both methods and no such difference could be detected (see Table 2). With respect to coverage, the results were nearly identical regarding the total number of peptides quantified.

## 3. Discussion

Technological advances in high-throughput ‘omics technologies have revolutionized scientific research and led to a significant increase in quantitative data output [31]. Metabolomics studies provide highly useful insights to many disciplines, from plant biology, to biomedical sciences, to environmental research. However, the diversity of metabolites’ physiochemical characteristics creates challenges in achieving broad metabolite recovery. For this, a variety of extraction techniques have been used to maximize metabolite recovery, including single-phase extraction—such as the commonly-employed MeOH-only precipitation method tested herein—which yields metabolites in a single supernatant layer, and biphasic methods—such as Matyash [32], Bligh and Dyer or Folch protocols [19,33], which use a mixture of methanol and chloroform—where each layer of two immiscible solvents will contain different metabolites that partition according to their polarity [34].

In comparison to single-phase extraction, the use of biphasic extraction allows for the extension of metabolite coverage to include additional metabolites with diverse physicochemical properties (SPE). However, there are many limitations to this method, one of them being that it is harder to automate than SPE due to multiple extraction steps and imprecise phase boundary locations. There are also many disadvantages specifically related to working with chloroform including toxicity, carcinogenicity, and it being hazardous to the environment. MS-based ‘omics that involve chloroform usage also necessitate the use of high-quality plastics and glassware with their associated higher cost, making it a less desirable choice of solvent overall, especially for high-throughput settings [35,36].

Consistent with previous findings regarding SPE methods, we observed in our study improved quantitative reproducibility for both metabolomic and proteomic data when using the MeOH-only extraction method as compared with the MeOH:CHCl_3_ method [37,38,39,40]. This reflects the simpler protocol with fewer manual handling steps and more consistent fraction collection, as there is no interphase of variable volume and shape that requires careful pipetting and which introduces a measure of irreproducibility to even the most carefully-carried-out protocol. Additionally, the lack of an interphase means the lack of interface boundary conditions where metabolite concentrations can vary greatly over short distances, which can in turn affect metabolite recovery. Quantifying samples from various biological sources did not reveal any increase in reproducibility with the MeOH-only technique over that of the MeOH:CHCl_3_ method, indicating that the inherent biological variation in these samples exceeds that of the laboratory handling for both protocols. Together, these observations suggest that when greater precision in quantitation is necessary, or where more statistical power is needed to investigate smaller effect sizes (fold-changes in metabolite abundance) than available sample numbers, an MeOH-only protocol may be more appropriate. However, as the identified metabolome coverage was greater when using the MeOH:CHCl_3_ method, and as biological variation can exceed the technical variation, where maximum metabolite coverage is required it may be appropriate to sacrifice the benefit in technical reproducibility and use the MeOH:CHCl_3_ method. In either case, for high-throughput facilities where automation is key, the difference in coverage, at least for identifiable metabolites, is sufficiently small that the MeOH-only extraction should be favored. 

Synergistic, multi-omics studies have the advantage of offering a combined perspective that facilitates the identification of biomarkers and mechanistic insights at several biological levels. This ultimately enhances disease diagnoses, monitoring, and the success of targeted therapies [41]. Here, our results suggest that where a joint proteomics-metabolomics study is undertaken the MeOH-only extraction/precipitation shows net benefit for the proteomics analysis, with no clear effect on peptide or protein coverage (see Appendix A), while yielding strong evidence of reduced quantitative variation at both the peptide and protein levels. Interestingly, this benefit was not observed when restricting the analysis to the results of the MaxLFQ protein quantitation algorithm available via the MaxQuant data processing software. As reported previously, the algorithm operates by delayed normalisation, which operates on the assumption of minimizing variance across and within samples [42]. Our results here validate this approach inasmuch as the algorithm appears to succeed in minimising protocol-associated variance. 

Thus, it appears that MeOH:CHCl_3_ extraction should be the method of choice for joint-omic blood plasma analyses, where the focus is on achieving the broadest metabolomic coverage possible and where automation is not required; in contrast, where quantitative proteomics may be the focus, MeOH precipitation should instead be selected.

## 4. Materials and Methods

### 4.1. Reagents

Acetonitrile (ACN), methanol (≥99.9%), chloroform, LC-MS CHROMASOLV, and deionized water were bought from Honeywell (Wunstorfer, Strasse, Seelze, Germany). Hydrochloric acid (HCl) (37%) was bought from VWR Chemicals (France). Formic acid (FA) and trifluoroacetic acid (TFA) were purchased from Fisher Scientific (Loughborough, UK). 

C18 columns, Pierce trypsin protease, lysis buffer, Pierce protease inhibitor tablets and lysyl-endopeptidase LysC were purchased from Thermo Scientific (Rockford, IL, USA). Bovine serum albumin and Bradford’s reagent were purchased from Sigma-Aldrich (St. Louis, MO, USA).

### 4.2. Samples Collection

Human whole blood samples (5 mL) were collected from healthy males and females (n = 6) via venipuncture into EDTA tube at University Hospital Sharjah. All participants gave informed consent for inclusion before enrollment in the study. Blood was centrifuged (14,000 rpm for 15 min at 24 °C) to separate plasma samples, which were stored at −80 °C.

### 4.3. Sample Preparation

#### 4.3.1. Methanol Precipitation for Metabolomic Extraction

An amount of 100 µL from each plasma sample was transferred to Eppendorf, then a volume of 300 µL of methanol was added to the aliquot. Samples were vortexed, then chilled at −20 °C for 2 h followed by centrifugation (14,000 rpm, 15 min, 24 °C) to precipitate protein. The metabolite-containing supernatants were collected and transferred to glass vials for drying using the EZ-2 Plus (GeneVac, Ipswich, UK) at 40 ± 1 °C and the protein pellets that remained were air-dried for proteomics (see Section 4.3.3). Dried metabolite samples were resuspended with 200 µL (0.1% formic acid in water), and vortexed for 2 min. The samples were filtered using a hydrophilic nylon syringe filter (0.45 μm pore size) and placed within glass inserts prior to being analyzed by Q-TOF MS.

#### 4.3.2. Methanol: Chloroform for Metabolomic Extraction

An amount of 100 µL from each sample was transferred to Eppendorf, then a volume of 400 µL of methanol and 300 µL of chloroform was added to the aliquot. Samples were vortexed, then centrifuged (14,000 rpm, 15 min, 24 °C). Two metabolite-containing layers, an upper aqueous and lower organic phase, were obtained, separated by a thin white proteinaceous disc. The upper layer for each sample was transferred to glass vials and a volume of 400 µL of methanol was added, followed by vortexing and centrifugation to pellet the protein disc, after which the remaining supernatant was transferred to the same glass vials as before for the drying step and the protein pellets that remained were air-dried for proteomics. Dried metabolomics samples were resuspended with 200 µL (0.1% formic acid in water) to be injected into HPLC and analyzed by Q-TOF MS.

#### 4.3.3. Proteomics Sample Preparation

The protein pellets were air-dried and resuspended in 100 µL denaturation buffer (6M urea and 2M thiourea in 10 mM Tris buffer at pH 8), after which proteins were quantified by modified Bradford [43].

Protein digestion and desalting. Protein samples measuring 100 µg were reduced using dithiothreitol (DTT) at a concentration of 1 mM, then incubated for 1 h with mild agitation at 100 rpm at room temperature (RT). Then, the samples were alkylated by adding 5.5 mM iodoacetamide (IAA), followed by incubation in the dark for 1 h at 100 rpm and RT. At the end of each step, the pH was measured and adjusted to 8.0 if necessary. After that, 1 µg of lysyl-endopeptidase LysC (1:100, *w*/*w*) was added, and samples were incubated for 3 h at 100 rpm at RT. Then, samples were diluted four times using 20 mM ammonium bicarbonate and digested with 1 µg trypsin (1:100 ratio) followed by incubation overnight at 100 rpm at RT prior to stopping the reaction by acidification to 1% TFA. 

In the desalting step, all the samples were dried then resuspended by 1% trifluoroacetic acid (TFA) and filtered using C18 STAGE (Stop And Go Extraction) tips. The conditioning of C18 membranes was performed by aspirating and dispensing 100 µL 50% ACN, followed by equilibration by 100 µL of 0.1% TFA. The protein samples were loaded by aspirating and dispensing the samples 10 times, and the columns were then washed using 0.1% TFA in 5% ACN. After this, peptides were eluted into LC vials by aspiration and dispensing 100 µL of 0.1% formic acid (FA) in 60% ACN. Subsequently, the eluted peptide samples were dried using EZ-2 Plus (GeneVac, Ipswich, UK), and then resuspended in 200 µL of 0.1% FA in 2% ACN prior to LC-MS/MS analysis. 

### 4.4. Liquid Chromatography Tandem Mass Spectrometry (LC-MS/MS)

Metabolomics analysis. The Elute UHPLC and Q-TOF Mass Spectrometer (Bruker, Bremen, Germany) were utilized for metabolites detection. The Elute HPG 1300 pumps, Elute Autosampler (Bruker, Bremen, Germany) and Hamilton^®^ Intensity Solo 2 C18 column (100 mm × 2.1 mm, 1.8 m beads) were employed using reversed-phase chromatography. Solvents used for separation were 0.1% FA in LC grade water (solvent A) and 0.1% FA in ACN (solvent B). Each metabolite and protein extract were analyzed in duplicate. 

The column was kept at 35 °C, and each sample was injected twice with an injection volume of 10 µL. Sample elution was performed by 30 min gradient, starting with 1% ACN for 2 min and then ramped to 99% ACN within 15 min. After that, 99 % ACN was kept for 3 min, and then the re-equilibration to 1% ACN was done for 10 min. The flow rate was 0.25 mL/min for 20 min, then 0.35 mL/min for 8.3 min, and then the flow rate was set at 0.25 mL/min for 1.7 min.

The ESI source conditions for every injection were as follows: the drying gas flow rate was 10.0 L/min at a temp of 220 °C; the capillary voltage was set at 4500 V; the End Plate offset was set at 500 V; the nebulizer pressure was 2.2 bar. 

For MS2 acquisition in metabolomics analysis, the collision energy was fluctuated between 100–250% of 20 eV and end plate offset of 500 V. The acquisition was in two sections: auto MS scan for the calibrant sodium formate in 0–0.3 min, and auto MS/MS for fragmentation, in 0.3 to 30 min. The instrument was operated in positive mode at 12 Hz for both acquisition sections. The scan range was from 20 to 1300 m/z, the precursor ion isolation width was set to ±0.5 m/z, the number of precursors selected for fragmentation was 3, the cycle time was 0.5 s, and the intensity threshold to 1000 counts. After three spectra, active exclusion was performed and released after 0.2 min. 

Proteomics analysis. A nano elute (Bruker Daltonics) coupled to a quadrupole-time-of-flight mass spectrometer (Q-TOF) was utilized to perform the LC-MS/MS analysis (Bruker Daltonics) with a CaptiveSpray ion source (Bruker Daltonics). An amount of the sample measuring 4 µL (4ug of the peptides) was injected and the separation was performed on a FIFTEEN column C18 15 cm × 75 um, 1.9 um (Bruker), using solvent A (0.1% formic acid in deionized water) and solvent B (0.1% formic acid in acetonitrile) with a 140-min gradient as follows: 0 to 5 min, 5% B; 5 to 120 min, 5–35% B; 120 to 125 min, 35–95% B; 125 to 135 min, 95% B; 135 to 135.2 min, 95–5% B; 135.2–140, 5% B. The flow rate was 0.30 µL/min. The CaptiveSpray ion source conditions for every injection were as follows: the drying gas flow rate was 3.0 L/min at a temp of 150 °C and the capillary voltage was set at 1600 V. The acquisition involved an auto MS/MS scan with CID acquisition, which included fragmentation. The acquisition was performed using the positive mode at 2 Hz. The automatic in-run mass scan range was from 150 to 2200 m/z, the width of the precursor ion was ±0.5, the number of precursors was 2, the cycle time was 3.0 s, and the threshold was 500 cts. Active exclusion was triggered after 1 spectrum and released after 2.0 min. For MS2 acquisition the data-dependent acquisition (DDA) was used, and the collision energy was adjusted between 23–65 eV as a function of the m/z value.

### 4.5. Data Analysis and Statistical Approach

Metabolomics Data Processing. For metabolomic analysis, MetaboScape^®^ 4.0 program (Bruker Daltonics, Billerica, MA, USA) was employed for data processing, feature extraction and metabolite identification. The T-ReX 2D/3D workflow was used to identify the molecular features with the following settings. The minimum peak length was set to 7 spectra and the minimum intensity threshold was 1000 counts for peaks detection. The peak area was employed for quantification and the injected external calibrant in the interval of 0–0.3 min was used to recalibrate the mass spectra. The selected mass to charge ratio (m/z) and retention time for scanning were in the ranges of 20–1300 m/z and 0.3–30 min, respectively. MS/MS spectra for features were averaged on import and features found in at least 12 of the 40 injections were taken into further consideration. Metabolites were identified by matching to the human metabolome database (HMDB) by combined MS/MS, precursor m/z values, and isotopic pattern scores. Where multiple features matched a given database entry, the annotation quality score (AQ score) was used to select only the best matching feature.

MetaboAnalyst (https://www.metaboanalyst.ca (accessed on 2 September 2022)) was used to perform volcano analyses to identify metabolites with significantly different abundance [*p*-value < 0.05 and log_2_(fold change) > 2]. Box plots were generated using GraphPad Prism 9 and Venn diagrams by using Bioinformatics and Evolutionary Genomics (http://bioinformatics.psb.ugent.be/webtools/Venn/ (accessed on 15 September 2022).

Proteomics Data Processing. To identify the peptides and proteins, the raw data were processed by MaxQuant [30] 1.6.17.0 using the UniProt proteome for Homo sapiens (Proteome ID: UP000005640, 79,052 entries, 14 April 2022) and the Andromeda search engine. Default parameters were applied in the MS/MS database search, including methionine oxidation and acetylation of protein N-termini as variable modifications and carbamidomethylation of cysteine residues as a fixed modification. Peptide spectral matching (PSM) was filtered with a 20-ppm precursor mass tolerance and 1% false discovery rate (FDR). For label-free quantification (LFQ), the MaxLFQ algorithm was used. In the in-silico digestion, the default trypsin/P enzymatic cleavage rule was utilized. To ensure a reliable label-free quantification, the protein groups were identified with at least two peptides assigned, and with at least one being unique to the protein.

## 5. Conclusions

We compared two commonly-employed extraction methods for simultaneous metabolomics and proteomics analyses in blood plasma, methanol:chloroform and methanol precipitation. This study revealed that MeOH:CHCl_3_ has higher metabolite recovery but similar proteomics recovery in comparison to MeOH-only, while the technical reproducibility of the MeOH precipitation was enhanced for both metabolomics and proteomics.

## Figures and Tables

**Figure 1 ijms-24-01354-f001:**
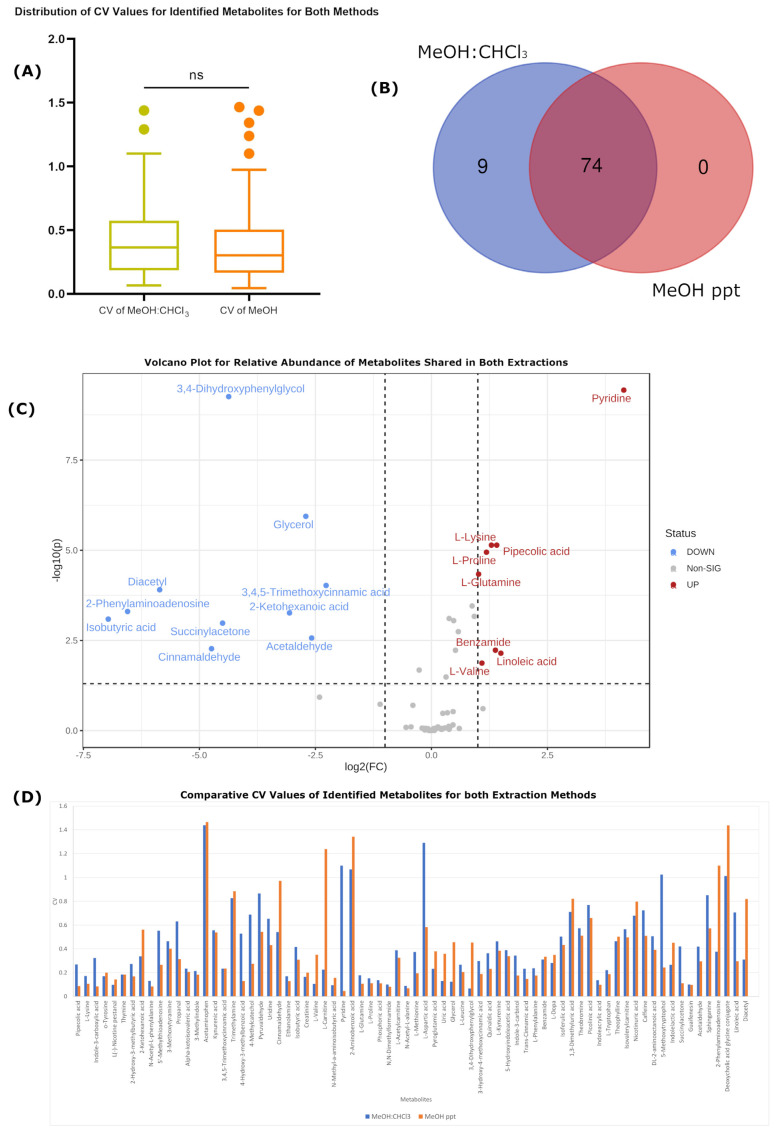
Representation of biological metabolomic variation through different statistical analyses. (**A**) Box plot that shows the distribution of CV values for identified metabolites for both methods. Yellow color represents CV values of metabolite intensities in MeOH:CHCl_3_, while orange colour represents CV values of metabolite intensities in MeOH-only. ns indicates not significant. (**B**) Venn diagram for the total number of metabolites found in each extraction method. (**C**) Volcano plot that presents the relative abundance of metabolites shared in both extraction methods. Red points indicate metabolites more abundant in methanol-only extracts, while blue points indicate metabolites more abundant in MeOH:CHCl_3_ extracts. Y-axis indicates −log10-transformed *p*-values (higher is more significant) and X-axis indicates log2-transformed fold change. (**D**) Column chart that presents the comparison of CV values within each metabolite between the two methods. Blue color represents CV values of metabolite intensities in MeOH:CHCl_3_, while orange colour represents CV values of metabolite intensities in MeOH-only.

**Figure 2 ijms-24-01354-f002:**
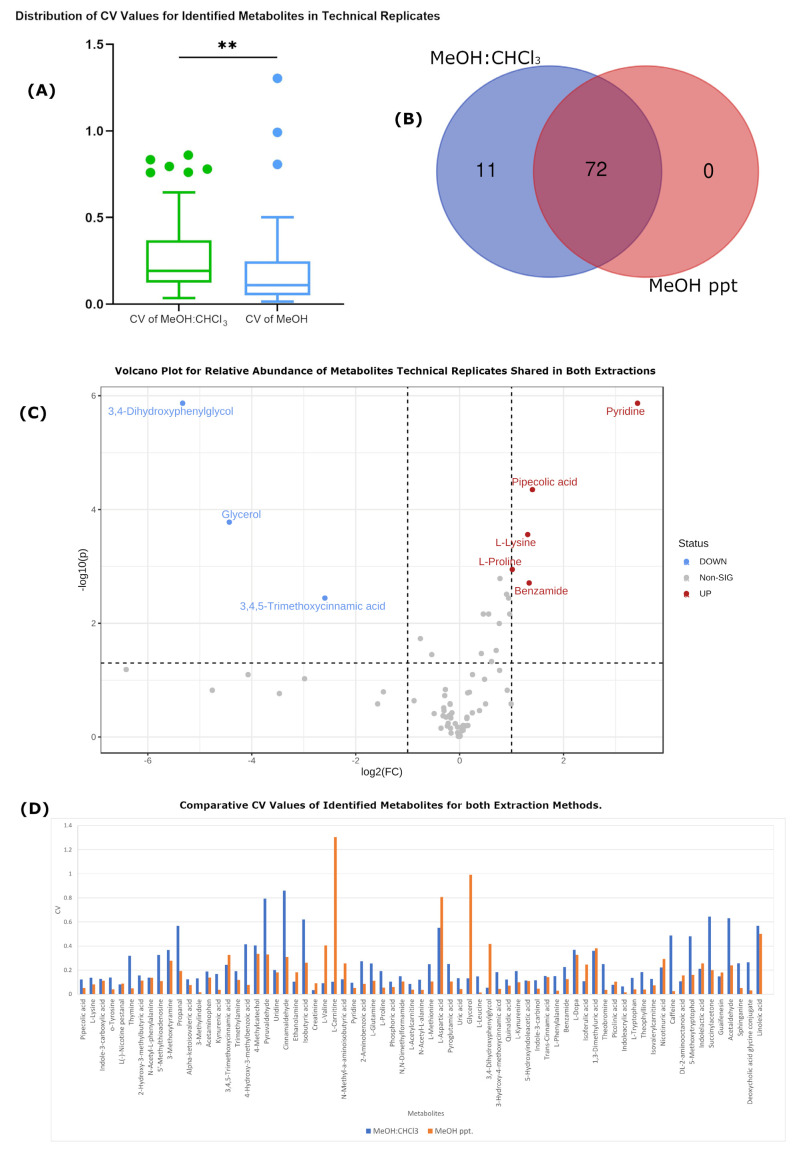
Representation of technical variation through different statistical analyses. (**A**) Box plot that shows the distribution of CV values for identified metabolites for both methods. Green color represents CV values of metabolite intensities in MeOH:CHCl_3_, while blue colour represents CV values of metabolites intensities in MeOH-only. ** indicates *p* ≤ 0.01 (**B**) Venn diagram for the total number of metabolites found in each extraction method. (**C**) Volcano plot that presents the relative abundance of metabolites shared in both extraction methods. Red color the metabolites of methanol-only, while blue color presents the metabolites of MeOH:CHCl_3_. (**D**) Column chart that presents the comparison of CV values within each metabolite between the two methods. Blue color represents CV values of metabolite intensities in MeOH:CHCl_3_, while orange colour represents CV values of metabolite intensities in MeOH-only.

**Figure 3 ijms-24-01354-f003:**
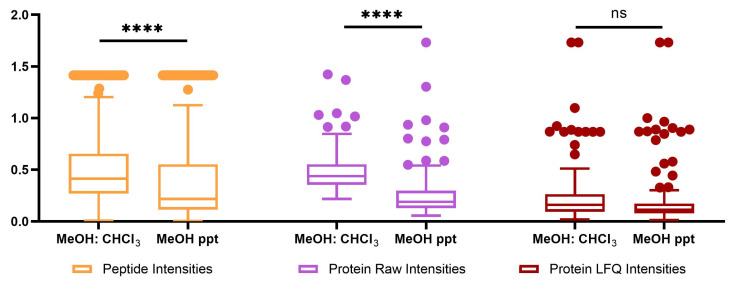
Analysis of variation for proteomics data: distribution of CV values for peptide, raw protein and LFQ protein intensities, compared between the two extraction methods. **** indicates *p* ≤ 0.0001, ns indicates not significant.

**Table 1 ijms-24-01354-t001:** Comparing numbers of identified metabolomics features and their average variation.

	MeOH: CHCl_3_		MeOH ppt	
	Different Biological Samples (*n* = 6)	Technical Replicates (*n* = 4)	Different Biological Samples (*n* = 6)	Technical Replicates (*n* = 4)
Number of detected features	5556	5702	3708	3584
Number of identified metabolites	83	83	74	72
Average RSD	0.427405	0.27521863	0.389594	0.179235

**Table 2 ijms-24-01354-t002:** Comparing numbers of identified proteins and their average variation.

	MeOH: CHCl_3_			MeOH ppt		
	Protein LFQ Intensities	Raw Protein Intensities	Peptides	Protein LFQ Intensities	Raw Protein Intensities	Peptides
Number of quantifiable features	104	114	1243	103	115	1254
Average RSD of quantitation	0.264476	0.482346	0.483063	0.232188	0.26544306	0.369035

## Data Availability

Proteomics mass spectrometry data have been deposited to the ProteomeXchange Consortium via the PRIDE partner repository with the data set identifier PXD038750. Metabolomics data are deposited in Metabolomics Workbench with study ID ST002391 (DOI: http://dx.doi.org/10.21228/M8WH85, accessed on 2 September 2022).

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
