# Peer review of "Evaluation of Two Simultaneous Metabolomic and Proteomic Extraction Protocols Assessed by Ultra-High-Performance Liquid Chromatography Tandem Mass Spectrometry"

_ijms, 2023, doi:10.3390/ijms24021354_

Round 1
Reviewer 1 Report
The manuscript entitled “Evaluation of Two Simultaneous Metabolomic and Proteomic Extraction Protocols Assessed by Ultra-High-Performance Liquid Chromatography-Tandem Mass Spectrometry” is generally well-written, the experimental design is optimal and could be accepted for publication. However, the following concerns should be addressed in the manuscript.
Comment 1. AuthorMeOH-only-based on the underlying chemistry (or possible underlying chemistry) driving MeOH-only based greater reproducibility for metabolomic and proteomic quantifications over MeOH: CHCl3 based approach. As this is a major finding, it becomes important to be supported by compelling chemical rationales and supported by relevant references.
Comment 2. Line 225-227 “Quantifying samples from various biological sources did not reveal any increase in reproducibility over the MeOH-only technique, indicating that the inherent biological variation in these samples exceeds that of the laboratory handling for both protocols”. This sentence is ambiguous. What kind of underlying biological variability might make the MeOH-only approach less reproducible?
Comment 3. Line 228-230 “Together, these observations suggest that when greater precision in quantization is necessary, or where the biological differences being investigated are small, a MeOH-only protocol may be more appropriate”. Here the authors remain unclear, about what the phrase “biological differences being investigated are small” means, the term small is undefined, is it a small number of samples, replicates, or investigations? Further, there should be some near numerical approximation if stating
any quantity. (per se, small may refer to ~2-3, or large as ~ 80-100).
Comment 4. It is convincing that the MeOH-only technique holds good for technical replicates, however, can it be as convincing for biological replicates as well?
Author Response
To the editor:
We wish to thank the editor and reviewers for their efforts and suggestions for improving the manuscript. We have edited the manuscript in accordance with the suggestions and comments made and believe the manuscript is better for these contributions. Our responses to the specific points made by the reviewers follow:
Reviewer 1
We wish to thank the reviewer for taking the time to read the manuscript and offer their comments and suggestions. With regards to their specific recommendations our responses are as follow:
Comment 1. Author MeOH-only-based on the underlying chemistry (or possible underlying chemistry) driving MeOH-only based greater reproducibility for metabolomic and proteomic quantifications over MeOH: CHCl3 based approach. As this is a major finding, it becomes important to be supported by compelling chemical rationales and supported by relevant references.
- We thank the reviewer for this question. As discussed in lines 229 to 235, we suggest that the basis of the improved reproducibility observed with MeOH-only extraction lies in the maintenance of all metabolites in a single phase, as contrasted with the biphasic solution formed with MeOH:CHCl3 extraction. With multiphasic systems boundary, layer conditions exist wherein differential metabolite concentrations can develop even with precise liquid handling by the operator. Additionally, such precise liquid handling is difficult to achieve even with careful manual manipulation by the operator, and such systems are often difficult to automate. Thus, there is an inevitable loss of quantitative reproducibility with biphasic extractions when compared with single phase extractions. We thus propose that it is the number of phases formed in each system that is responsible for the observed differences in reproducibility, rather than the relative hydrophobicity or chaotropic effects, per se, of the different solvent systems used. This is in agreement with other studies examining single vs multiphasic systems and we have added references to the discussion (lines 229 to 235) accordingly. We have also edited the text to include interphasic boundary conditions in the discussion of the effect of the interphase on reproducibility.
Comment 2. Line 225-227 “Quantifying samples from various biological sources did not reveal any increase in reproducibility over the MeOH-only technique, indicating that the inherent biological variation in these samples exceeds that of the laboratory handling for both protocols”. This sentence is ambiguous. What kind of underlying biological variability might make the MeOH-only approach less reproducible?
- Biological variability can be due to normal physiological processes and differences inherent to the samples, such as age, diet, medication, physical activity, genetic background, and demographic diversity. The variability observed in measurements between samples is a compound of the inherent differences between samples and the added variability of the protocol by which they are processed. Thus, we do not argue that biologically inherent variation will decrease the MeOH-only extraction reproducibility, but rather that the existence of this variation in the samples can exceed that of, and hence hide, the technical variation related to the extraction techniques being compared.
Comment 3. Line 228-230 “Together, these observations suggest that when greater precision in quantization is necessary, or where the biological differences being investigated are small, a MeOH-only protocol may be more appropriate”. Here the authors remain unclear, about what the phrase “biological differences being investigated are small” means, the term small is undefined, is it a small number of samples, replicates, or investigations? Further, there should be some near numerical approximation if stating
any quantity. (per se, small may refer to ~2-3, or large as ~ 80-100).
- We thank the reviewer for this observation. They are indeed correct that the reference to “small” is ambiguous and no range is given. As the statistical power (1-β) for any given study to detect (at any given α) effect sizes of any specified magnitude is a function of the sample size and variability in the data, there is no predetermined notion of what “small” here should mean. Rather, it is meant that if all else is equal (i.e. sample sizes and accepted α), that the protocol with lower variability (MeOH-only) will give more statistical power for detecting smaller effect sizes (metabolite fold-changes) than would the alternate protocol.
- We have edited the text (see line 243) to make clearer to readers that fold-changes in metabolite abundance are the effect size to which is being referred, and to clarify that the meaning is in respect to improved statistical power for smaller effect sizes relative to that which could be detected with the alternate protocol.
Comment 4. It is convincing that the MeOH-only technique holds good for technical replicates, however, can it be as convincing for biological replicates as well?
- We appreciate this question. As the observed biological variability is a compound of inherent biological and technical variability, it can be assumed that with larger sample sizes to account for the inherent biological variation, differences in the technical reproducibility of the used protocols would ultimately become visible. While this cannot be shown with the data presented here, the enhanced technical reproducibility of the single-phase extraction method demonstrates the improved reproducibility of the protocol itself with plasma samples.
Reviewer 2 Report
In this work, Zenati et al. described the comparison of two commonly used methods for the extraction of metabolome and proteome from blood plasma samples by LC-MS/MS. The validation of technical variation of the two protocols is examined by incorporating biological variation and repeated analysis of aliquots from a single sample. The results show that methanol:chloroform extraction has higher metabolite recovery but similar proteomics recovery to methanol precipitation. The design and analysis of the experimental results are well described, and I believe just a few details would make the conclusion more reliable.
1. In section 2.1, the coefficient of variance is used to measure the quantitative variability of each method. Can the authors add one more plot to show the comparison of CV values within each metabolite between the two methods? The variability of different metabolites could have different range and I believe it would interesting to show if such effects exist.
2. The proteomics recovery is visualized as Venn diagram for the total number of metabolites 143 found in each extraction method. Could the author add one more plot two show the correlation between two extraction protocols in both metabolome and proteome that are identified in both methods? This would be a better way to validate the quality of the discovery.
Author Response
To the editor:
We wish to thank the editor and reviewers for their efforts and suggestions for improving the manuscript. We have edited the manuscript in accordance with the suggestions and comments made and believe the manuscript is better for these contributions. Our responses to the specific points made by the reviewers follow:
Reviewer 2
We wish to thank the reviewer for taking the time to read the manuscript and offer their comments and suggestions. With regards to their specific recommendations our responses are as follow:
Comments and Suggestions for Authors (second reviewer)
- In section 2.1, the coefficient of variance is used to measure the quantitative variability of each method. Can the authors add one more plot to show the comparison of CV values within each metabolite between the two methods? The variability of different metabolites could have different range and I believe it would be interesting to show if such effects exist.
- We thank the reviewer for this suggestion. We have added two additional figures (panel D in figures 1 and 2) showing the relative coefficient of variance (CV) values for each metabolite.
- The proteomics recovery is visualized as Venn diagram for the total number of metabolites 143 found in each extraction method. Could the author add one more plot two show the correlation between two extraction protocols in both metabolome and proteome that are identified in both methods? This would be a better way to validate the quality of the discovery.
- We thank the reviewer for this suggestion. We have included an additional panel in supplementary materials showing the overall correlation of the observed protein and metabolite abundances as requested and we have edited the text (see line 257).